# Revisiting Implicit Differentiation for Learning Problems in Optimal Control

**Ming Xu**
School of Computing
Australian National University
`mingda.xu@anu.edu.au`

**Timothy Molloy**
School of Engineering
Australian National University
`timothy.molloy@anu.edu.au`

**Stephen Gould**
School of Computing
Australian National University
`stephen.gould@anu.edu.au`

## Abstract

This paper proposes a new method for differentiating through optimal trajectories arising from non-convex, constrained discrete-time optimal control (COC) problems using the implicit function theorem (IFT). Previous works solve a differential Karush-Kuhn-Tucker (KKT) system for the trajectory derivative, and achieve this efficiently by solving an auxiliary Linear Quadratic Regulator (LQR) problem. In contrast, we directly evaluate the matrix equations which arise from applying variable elimination on the Lagrange multiplier terms in the (differential) KKT system. By appropriately accounting for the structure of the terms within the resulting equations, we show that the trajectory derivatives scale linearly with the number of timesteps. Furthermore, our approach allows for easy parallelization, significantly improved scalability with model size, direct computation of vector-Jacobian products and improved numerical stability compared to prior works. As an additional contribution, we unify prior works, addressing claims that computing trajectory derivatives using IFT scales quadratically with the number of timesteps. We evaluate our method on a both synthetic benchmark and four challenging, learning from demonstration benchmarks including a 6-DoF maneuvering quadrotor and 6-DoF rocket powered landing.

## 1 Introduction

This paper addresses end-to-end learning problems that arise in the context of constrained optimal control, including trajectory optimization, inverse optimal control, and system identification. We propose a novel, computationally efficient approach to computing analytical derivatives of state and control trajectories that solve constrained optimal control (COC) problems with respect to underlying parameters in cost functions, system dynamics, and constraints (e.g., state and control limits).

The efficient computation of these *trajectory derivatives* is important in the (open-loop) solution of optimal control problems for both trajectory optimization and model predictive control (MPC). Such derivatives are also crucial in inverse optimal control (also called inverse reinforcement learning or learning from demonstration), where given expert demonstration trajectories, the objective is to compute parameters of the cost function that best explain these trajectories. Extensions of inverse optimal control that involve inferring parameters of system dynamics in addition to cost functions also subsume system identification, and provide further motivation for the efficient computation of trajectory derivatives. Our proposed method of computing trajectory derivatives enables direct

37th Conference on Neural Information Processing Systems (NeurIPS 2023).

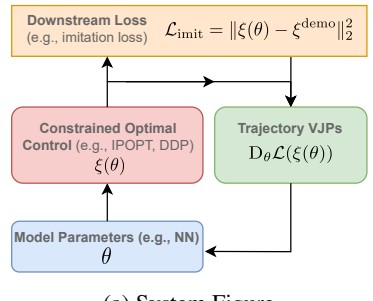
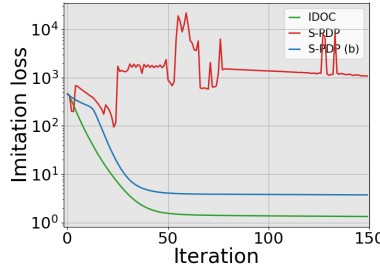

(a) System Figure

(b) Imitation Learning: Rocket Landing

Figure 1: **Left a):** Our approach addresses learning problems in optimal control such as imitation learning. IDOC provides a method for computing the trajectory derivative or alternatively, vector-Jacobian products with respect to an outer-level loss, important for solving this problem using an end-to-end learning approach. **Right b):** IDOC yields superior trajectory derivatives for the imitation learning task when inequality constraints (rocket tilt and thrust limits) are present.

minimization of a loss function defined over optimal trajectories (e.g., imitation loss) using first-order descent techniques and is a direct alternative to existing works [2, 20, 22], including those based on bespoke derivations of Pontryagin's maximum principle (PMP) in discrete time [20, 22].

Analytical trajectory derivatives for COC problems are derived by differentiating through the optimality conditions of the underlying optimization problem and applying Dini's implicit function theorem. Derivatives are recovered as the solution to a system of linear equations commonly referred to as the (differential) KKT system. This framework underlies all existing works [2, 20, 22] as well as our proposed approach. Similarly to Amos et al. [2], we construct the differential KKT system, however we use the identities from Gould et al. [14] that apply variable elimination on the Lagrange multipliers relating to the dynamics and constraints. We show how to exploit the block-sparse structure of the resultant matrix equations to achieve linear complexity in computing the trajectory derivative with trajectory length. Furthermore, we can parallelize the computation, yielding superior scalability and numerical stability on multithreaded systems compared to methods derived from the PMP [20, 22].

Our specific contributions are as follows. First, we derive the analytical trajectory derivatives for a broad class of constrained (discrete-time) optimal control problems with additive cost functions and dynamics described by first-order difference equations. Furthermore, we show that the computation of these derivatives is linear in trajectory length by exploiting sparsity in the resulting matrix expressions. Second, we describe how to parallelize the computation of the trajectory derivatives, yielding lower computation time and superior numerical stability for long trajectories. Third, we show how to directly compute vector-Jacobian products (VJPs) with respect to some outer-level loss over optimal trajectories, yielding further improvements to computation time in the context of bi-level optimization. This setting commonly arises in the direct solution of inverse optimal control problems [17, 33, 34]. Finally, we provide discussion unifying existing methods for computing trajectory derivatives [2, 20, 22]. We dub our method IDOC (Implicit Differentiation for Optimal Control) and validate it across numerous experiments, showing a consistent speedup over existing approaches derived from the PMP. Furthermore, for constrained problems, IDOC provides significantly improved trajectory derivatives, resulting in superior performance for a general learning from demonstration (LfD) task.[1].

## 2 Related Work

**Differentiating Through Optimal Control Problems.** An optimal control (OC) problem consists of (system) dynamics, a cost function, an optimal control policy, and a set of state and/or control constraints. Learning problems in optimal control involve learning some or all of these aspects, with the problem of learning dynamics referred to as system identification [29], the problem of learning the cost function referred to as inverse optimal control (or inverse reinforcement learning or learning from demonstration) [34, 17, 21, 33, 37, 51], and the problem of learning the control policy referred to as reinforcement learning [4]. Recent work [20] has shown that solving these

---

[1]Code available at https://github.com/mingu6/Implicit-Diff-Optimal-Control

learning problems can be approached in a unified end-to-end fashion by minimizing task-specific loss functions with respect to (unknown) parameters of the associated OC problem. This unified formulation places great importance on the efficient computation of derivatives of optimal trajectories with respect to said parameters. Methods reliant on computing trajectory derivatives have, however, been mostly avoided (and argued against) in the robotics and control literature due to concerns about computational tractability. For example, bi-level methods of inverse optimal control [34, 33] have mostly been argued against in favor of methods that avoid derivative calculations by instead seeking to satisfy optimality conditions derived from KKT [24, 12] or PMP conditions [17, 32, 23, 21].

**Analytical Trajectory Derivatives.** Our method is a direct alternative to methods such as DiffMPC [2], PDP [20] and its extension Safe-PDP [22], which differentiate through optimality conditions to derive trajectory derivatives. Common to these methods is identifying that trajectory derivatives can be computed by solving an auxiliary affine-quadratic OC problem, and furthermore, that this can be done efficiently using a matrix Riccati equation. While our approach still differentiates the optimality conditions, we avoid solving matrix Riccati equations and show that this enables easy computation of VJPs, parallelization across timesteps and improved numerical stability. Like Safe-PDP, IDOC computes derivatives through COC problems with smooth inequality constraints, however we show in our experiments that our derivatives are significantly more stable during training. Section 4.3 provides a detailed description of the differences between IDOC and existing methods.

**Differentiable Optimization.** Our work falls under the area of differentiable optimization, which aims to embed optimization problems within end-to-end learning frameworks. Gould et al. [14] proposed the deep declarative networks framework, and provide identities for differentiating through continuous, inequality-constrained optimization problems. Combinatorial problems [31, 44], as well as specialized algorithms for convex problems [1] have also been addressed. Differentiable optimization has been applied to end-to-end learning of state estimation [49, 42] and motion planning problems [6, 2, 26] in robotics, see Pineda et al. [35] for a recent survey. In addition, numerous machine learning and computer vision tasks such as pose estimation [9, 40], meta-learning [28], sorting [11, 7] and aligning time series [47] have been investigated under this setting.

# 3 Constrained Optimal Control Formulation

In this section, we provide an overview of the COC problems we consider in this paper. These problems involve minimizing a cost function with an additive structure, subject to constraints imposed by system dynamics and additional arbitrary constraints such as state and control limits. As a result, they can be formulated as a non-linear program (NLP). We will discuss how to differentiate through these COC problems in Section 4.

## 3.1 Preliminaries

First, we introduce notation around differentiating vector-valued functions with respect to vector arguments, consistent with Gould et al. [14]. Let $f : \mathbb{R}^n \to \mathbb{R}^m$ be a vector-valued function with vector arguments and let $\mathrm{D}f \in \mathbb{R}^{n \times m}$ be the (matrix-valued) derivative where elements are given by

$$(\mathrm{D}f(x))_{ij} = \frac{\partial f_i}{\partial x_j}(x). \tag{1}$$

For a scalar-valued function with (multiple) vector-valued inputs $f : \mathbb{R}^n \times \mathbb{R}^m \to \mathbb{R}$ evaluated as $f(x, y)$, let the second order derivatives $\mathrm{D}^2_{XY} = \mathrm{D}_X(\mathrm{D}_Y f)^\top$. See Gould et al. [14] for more details.

## 3.2 Optimization Formulation for Constrained Optimal Control

We can formulate the (discrete-time) COC problem as finding a cost-minimizing trajectory subject to constraints imposed by (possibly non-linear) system dynamics. In addition, trajectories may be subject to further constraints such as state and control limits.

To begin, let the dynamics governing the COC system at time $t$ be given by $x_{t+1} = f_t(x_t, u_t; \theta)$, where $x_t \in \mathbb{R}^n$, $u_t \in \mathbb{R}^m$ denotes state and control variables, respectively[2]. For time horizon $T > 0$,

---

[2]The subscript $f_t$ allows for time-varying dynamics in principle, though we do not evaluate IDOC on any optimal control problems that have this property in our experiments.

let $x \triangleq (x_0, x_1, \ldots, x_T)$ and $u \triangleq (u_0, u_1, \ldots, u_{T-1})$ denote the state and control trajectories, respectively. Furthermore, let $\theta \in \mathbb{R}^d$ be the parameter vector that parameterizes our COC problem. Our COC problem is then equivalent to the following constrained optimization problem

$$
\begin{aligned}
&\text{minimize} \quad J(x, u; \theta) \triangleq \sum_{t=0}^{T-1} c_t(x_t, u_t; \theta) + c_T(x_T; \theta) \\
&\text{subject to}
\end{aligned}
$$

$$
\begin{array}{lll}
x_0 = x_{\text{init}} & & \text{(initial state)} \\
x_{t+1} - f_t(x_t, u_t; \theta) = 0 & \forall t \in \{0, \ldots, T-1\} & \text{(dynamics)} \\
g_t(x_t, u_t; \theta) \leq 0 & \forall t \in \{0, \ldots, T-1\}, & \text{(path ineq. constraints)} \\
h_t(x_t, u_t; \theta) = 0 & \forall t \in \{0, \ldots, T-1\}, & \text{(path eq. constraints)} \\
g_T(x_T; \theta) \leq 0 & & \text{(terminal ineq. constraints)} \\
h_T(x_T; \theta) = 0, & & \text{(terminal eq. constraints)}
\end{array} \tag{2}
$$

where $f_t : \mathbb{R}^n \times \mathbb{R}^m \times \mathbb{R}^d \to \mathbb{R}^n$ describe the dynamics, and $c_t : \mathbb{R}^n \times \mathbb{R}^m \times \mathbb{R}^d \to \mathbb{R}$, $c_T : \mathbb{R}^n \times \mathbb{R}^d \to \mathbb{R}$ are the instantaneous and terminal costs, respectively. Furthermore, the COC system may be subject to (vector-valued) inequality constraints $g_t : \mathbb{R}^n \times \mathbb{R}^m \times \mathbb{R}^d \to \mathbb{R}^{q_t}$ and $g_T : \mathbb{R}^n \times \mathbb{R}^d \to \mathbb{R}^{q_T}$ such as control limits and state constraints, for example. Additional equality constraints $h_t : \mathbb{R}^n \times \mathbb{R}^m \times \mathbb{R}^d \to \mathbb{R}^{s_t}$ and $h_T : \mathbb{R}^n \times \mathbb{R}^d \to \mathbb{R}^{s_T}$ can also be included.

The decision variables of Equation 2 are the state and control trajectory $\xi \triangleq (x, u)$, whereas parameters $\theta$ are assumed to be fixed. Equation 2 can be interpreted as an inequality-constrained optimization problem with an additive cost structure and vector-valued constraints for the initial state $x_0$, dynamics, etc. over all timesteps. Let $\xi(\theta) \triangleq (x^\star, u^\star)$ be an optimal solution to Equation 2. We can treat the optimal trajectory $\xi(\theta)$ as an *implicit* function of parameters $\theta$, since Equation 2 can be solved to yield an optimal $\xi(\theta)$ for any (valid) $\theta$.

We can solve Equation 2 for the optimal trajectory $\xi(\theta)$ using a number of techniques. We can use general purpose solvers [13, 45], as well as specialized solvers designed to exploit the structure of COC problems, e.g., ones based on differential dynamic programming [30, 19]. Regardless, for the purposes of computing analytical trajectory derivatives using our method (as well as methods that differentiate through optimality conditions [20, 22, 2]), we only need to ensure that our solver returns a vector $\xi(\theta)$ which is a (local) minimizer to Equation 2. We now describe how to differentiate through these optimal control problems, important in the end-to-end learning context.

# 4 Trajectory Derivatives using Implicit Differentiation

In this section, we present our identities for computing trajectory derivatives $\mathrm{D}\xi(\theta)$ based on those derived in Gould et al. [14] for general optimization problems by leveraging first-order optimality conditions and the implicit function theorem. We exploit the block structure of the matrices in these identities that arises in optimal control problems to enable efficient computation and furthermore, show that computing trajectory derivatives is linear in the trajectory length $T$.

Before we present the main analytical result, recall the motivation for computing $\mathrm{D}\xi(\theta)$. Suppose in the LfD context we have a demonstration trajectory $\xi^{\text{demo}}$ (we can extend this to multiple trajectories, but choose not to for notational simplicity). Furthermore, define a loss $\mathcal{L}(\xi(\theta), \xi^{\text{demo}})$ which measures the deviation from predicted trajectory $\xi(\theta)$ to demonstration trajectory $\xi^{\text{demo}}$. Ultimately, if we wish to minimize the loss $\mathcal{L}$ with respect to parameters $\theta$ using a first-order decent method, we need to compute $\mathrm{D}_\theta \mathcal{L}(\xi(\theta), \xi^{\text{demo}})$ by applying the chain rule. Specifically, we compute $\mathrm{D}_\theta \mathcal{L}(\xi) = \mathrm{D}_\xi \mathcal{L}(\xi) \mathrm{D}\xi(\theta)$, which requires trajectory derivative $\mathrm{D}\xi(\theta)$.

## 4.1 Preliminaries

First, we first reorder $\xi$ such that $\xi = (x_0, u_0, x_1, u_1, \ldots, x_T) \in \mathbb{R}^{n_\xi \times 1}$, where $n_\xi = (n + m)T + n$. This grouping of decision variable blocks w.r.t. timesteps is essential for showing linear time complexity of the computation of the derivative. Let $\xi_t \in \mathbb{R}^{n+m}$ represent the subset of variables in $\xi$ associated to time $t$, with final state $\xi_T = x_T \in \mathbb{R}^n$. Next, we stack all constraints defined in Equation 2 into a single vector-valued constraint $r$ comprised of $T + 2$ blocks. Specifically, let $r(\xi; \theta) \triangleq (r_{-1}, r_0, r_1, \ldots, r_T) \in \mathbb{R}^{n_r}$, where $n_r = (T + 1)n + s + q$. The first block is given by

$r_{-1} = x_0$, and subsequent blocks are given by

$$r_t = \begin{cases} (\tilde{g}_t(x_t, u_t; \theta), h_t(x_t, u_t; \theta), x_{t+1} - f(x_t, u_t; \theta)) & \text{for } 0 \le t \le T - 1 \\ (\tilde{g}_T(x_T; \theta), h_T(x_T; \theta)) & \text{for } t = T. \end{cases} \tag{3}$$

Here, $\tilde{g}_t$ are the subset of *active* inequality constraints for $\xi$ (detected numerically using a threshold $\epsilon$) at timestep $t$. Note, $s = \sum_{t=0}^{T} s_t$, $q = \sum_{t=0}^{T} |\tilde{g}_t|$ are the total number of additional equality and active inequality constraints (on top of the $Tn$ dynamics and $n$ initial state constraints). Each block represents a group of constraints associated with a particular timestep (except the first block $r_{-1}$). As we will see shortly, grouping decision variables and constraints in this way will admit a favorable block-sparse matrix structure for cost/constraint Jacobians and Hessians required to compute $D\xi(\theta)$.

### 4.2 Analytical Results for Trajectory Derivatives

**Proposition 1** (IDOC). *Consider the optimization problem defined in Equation 2. Suppose $\xi(\theta)$ exists which minimizes Equation 2. Furthermore, assume $f_t$, $c_t$, $g_t$ and $h_t$ are twice differentiable for all $t$ in a neighborhood of $(\theta, \xi)$. If $\mathrm{rank}(A) = n_r$ and furthermore, $H$ is non-singular, then*

$$D\xi(\theta) = H^{-1}A^{\top}(AH^{-1}A^{\top})^{-1}(AH^{-1}B - C) - H^{-1}B, \tag{4}$$

*where*

$$A = D_\xi r(\xi; \theta) \in \mathbb{R}^{n_r \times n_\xi}$$

$$B = D_{\theta\xi}^2 J(\xi; \theta) - \sum_{t=-1}^{T} \sum_{i=1}^{|r_t|} \lambda_{t,i} D_{\theta\xi}^2 r_t(\xi; \theta)_i \in \mathbb{R}^{n_\xi \times d}$$

$$C = D_\theta r(\xi; \theta) \in \mathbb{R}^{n_r \times d}$$

$$H = D_{\xi\xi}^2 J(\xi; \theta) - \sum_{t=-1}^{T} \sum_{i=1}^{|r_t|} \lambda_{t,i} D_{\xi\xi}^2 r_t(\xi; \theta)_i \in \mathbb{R}^{n_\xi \times n_\xi},$$

*and Lagrange multipliers $\lambda \in \mathbb{R}^{n_r}$ satisfies $\lambda^{\top} A = D_\xi J(\xi; \theta)$.*

*Proof.* This is a direct application of Proposition 4.5 in Gould et al. [14]. $\qquad \square$

**Remark 1.** *Matrix $H$ is block diagonal with $T + 1$ blocks.*

To see this, $\hat{H} = D_{\xi\xi}^2 J(\xi; \theta)$ has a block diagonal structure with blocks given by $D_{\xi_t \xi_t}^2 c_t(\xi_t; \theta)$, which follows from the assumption of additive costs in the COC problem described in Equation 2. In addition, constraints $g_t$, $h_t$ depend only on $\xi_t$ and furthermore, the dynamics constraint is first order in $\xi_{t+1}$. Therefore, $D_{\xi\xi}^2 r_t(\xi; \theta)_i$ is only non-zero in the block relating to $\xi_t$ for all $t$ and $i$. We plot the block-sparsity structure of $H$ in Figure 2a.

**Remark 2.** *Matrix $A$ is a block-banded matrix that is two blocks wide.*

This is shown by noting that the only non-zero blocks in $A$ correspond to $D_{\xi_t} r_t(\xi; \theta)$ and $D_{\xi_{t+1}} r_t(\xi; \theta)$. The former relates to $g_t$, $h_t$ and the $-f_t(\xi_t; \theta)$ component of the dynamics constraint. The latter only relates to the $x_{t+1}$ component of the dynamics constraint. Finally, the initial condition block $r_{-1}$ only depends on $\xi_0$, hence the only non-zero block is given by $D_{\xi_0} r_{-1}(\xi_0; \theta)$. We plot the block-sparsity structure of $A$ in Figure 2b.

**Proposition 2.** *Evaluating Equation 4 has $O(T)$ time complexity.*

*Proof.* From Remarks 1 and 2, $H$ and $A$ have a block diagonal and block-banded structure with $T + 1$ and $2T + 2$ blocks, respectively. Therefore, we can evaluate $H^{-1}A^{\top}$ in $O(T)$ time and furthermore, $H^{-1}A^{\top}$ has the same structure as $A$. It follows that we can also evaluate $AH^{-1}B - C$ in $O(T)$ time after partitioning $B$ and $C$ into blocks based on groupings of $\xi$ and $r$.

Next, observe that $AH^{-1}A^{\top}$ yields a block tridiagonal matrix with $T + 2$ blocks on the main diagonal (see Figure 2c for a visualization of the block-sparse structure). As a result, we can evaluate $(AH^{-1}A^{\top})^{-1}(AH^{-1}B)$ by solving the linear system $(AH^{-1}A^{\top})Y = AH^{-1}B$ for $Y$ in $O(T)$

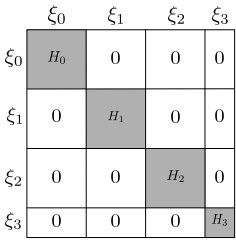

(a) Matrices $H$ and $H^{-1}$

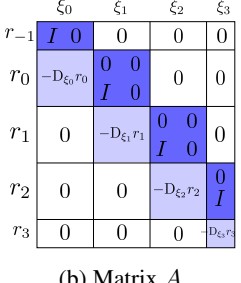

(b) Matrix $A$

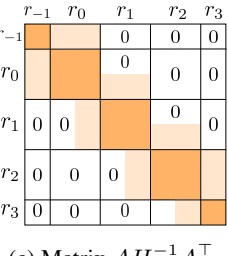

(c) Matrix $AH^{-1}A^\top$

Figure 2: Block-sparse structure for matrices $H$, $A$, and $AH^{-1}A^\top$ assuming $T = 3$. For $A$, we combined inequality and equality constraint groups $g_t$ and $h_t$ into a single group for simplicity. Shaded regions indicate (possible) non-zero blocks.

time using any linear time (w.r.t. number of diagonal blocks) block tridiagonal linear solver. A simple example is the block tridiagonal matrix algorithm, easily derived from block Gaussian elimination.

Finally, $H^{-1}B$ can easily be evaluated in $O(T)$ time by solving $T + 1$ linear systems comprised of the blocks of $H$ and $B$. We conclude that evaluating Equation 4 requires $O(T)$ operations. $\qquad\square$

### 4.3 Comparison to DiffMPC and PDP

All existing methods for computing analytical trajectory derivatives involve solving the linear system

$$\underbrace{\begin{bmatrix} H & A^\top \\ A & 0 \end{bmatrix}}_{K} \begin{bmatrix} \mathrm{D}\xi \\ -\mathrm{D}\lambda \end{bmatrix} = \begin{bmatrix} -B \\ -C \end{bmatrix}, \tag{5}$$

where blocks $H, A, B, C$ are defined in Section 4.2 and $\mathrm{D}\xi$ is the desired trajectory derivative. We call Equation 5 the (differential) *KKT system* and the matrix $K$ the (differential) *KKT matrix*. Note, $\mathrm{D}\xi$ is unique if $K$ is non-singular[3]. To better contextualize IDOC, we now briefly compare and contrast how previous methods such as DiffMPC [2] and PDP [20, 22] solve this KKT system.

**DiffMPC.** DiffMPC [2] proposes a method for differentiating through Linear Quadratic Regulator (LQR) problems, which are OC problems where the cost function is (convex) quadratic and the dynamics are affine. The authors show that solving the LQR problem using matrix Riccati equations can be viewed as an efficient method for solving a KKT system which encodes the optimality conditions. In addition, they show that computing trajectory derivatives involves solving a similar KKT system, motivating efficient computation by solving an auxiliary LQR problem in the backward pass. The matrix equation interpretation of the backward pass allows the efficient computation of VJPs for some downstream loss $\mathcal{L}(\xi)$ in a bi-level optimization context.

DiffMPC extends to handling non-convex OC problems with box constraints on control inputs via an iterative LQR-based approach. Specifically, such problems are handled by first computing VJPs w.r.t. the parameters of an LQR approximation to the non-convex problem around the optimal trajectory $\xi, \lambda$. Next, the parameters of the approximation are differentiated w.r.t. the underlying parameters $\theta$ and combined using the chain rule. However, differentiating the quadratic cost approximation and dynamics requires evaluating costly higher-order derivatives, e.g., $\frac{d}{d\theta}\frac{d^2 c(\xi)}{d\xi^2}$, which are 3D tensors. These tensors are dense in general, although problem specific sparsity structures may exist.

**PDP/Safe-PDP.** PDP [20] and its extension, Safe-PDP [22], take a similar approach to DiffMPC in deriving trajectory derivatives. PDP derives trajectory derivatives by starting with PMP, which is well-known in the control community and applies to non-convex OC problems. Furthermore, Jin et al. [20] shows that the PMP and KKT conditions are equivalent in the discrete-time COC setting. Trajectory derivatives are obtained by differentiating the PMP conditions, yielding a new set of PMP conditions for an auxiliary LQR problem, which can be solved using matrix Riccati equations. This

---

[3]See Section 10.1 in [8] for further discussion on conditions for non-singular $K$

approach is fundamentally identical to DiffMPC, with only superficial differences for LQR problems (solving a differential KKT system versus differentiating PMP conditions)[4].

For non-convex problems, PDP does not evaluate higher-order derivatives, unlike DiffMPC. However, it is not clear how to compute VJPs under approaches derived using the PDP framework. Safe-PDP extended PDP to handle arbitrary, smooth inequality constraints using the constrained PMP conditions, which extends the capability of DiffMPC to handle box constraints. Trajectory derivatives are computed by solving an auxiliary equality constrained LQR problem. Experiments in both Jin et al. [22] and Section 6 show that Safe-PDP yields unreliable derivatives for COC problems. While Jin et al. [22] conjected that instability was due to the set of active constraints changing between iterations, we find that IDOC still learns reliably in the presence of constraint switching. We instead observed that the poor gradient quality arises naturally from the equality constrained LQR solver used for the backward pass [25] not matching the solution obtained by directly solving the KKT system.

**IDOC.** In contrast, we solve Equation 5 for $D\xi$ by first applying variable elimination on $D\lambda$, yielding Proposition 1. We will discuss in Section 5 how the equations resulting from this approach admit parallelization and favorable numerical stability compared to DiffMPC and PDP's auxiliary LQR approach. Furthermore, VJPs can be easily computed (unlike PDP) without evaluating higher-order derivatives (unlike DiffMPC). IDOC handles arbitrary smooth constraints like Safe-PDP, and we show in Section 6 that our derivatives for COC problems are reliable during training. However, we require the additional assumption that $H$ is non-singular (which holds if and only if all blocks in $H$ are non-singular), which is not required for $K$ to be non-singular. An oftentimes effective solution when $H$ is singular, initially proposed by Russell et al. [39], is to set $H = H + \frac{\delta}{2}I$ for small $\delta$, which is analogous to adding a proximal term to the cost function in Equation 2. We will now describe how to evaluate Equation 4 in linear time by exploiting the block-sparse structure of the matrix equation.

# 5 Algorithmic Implications of IDOC

## 5.1 Parallelization and Numerical Stability

**Parallelization.** To evaluate Equation 4, we can leverage the block diagonal structure of $H$ and block-banded structure of $A$ to compute $H^{-1}A^\top$ and $H^{-1}B$ in parallel across all timesteps. Specifically, we evaluate the matrix product block-wise, e.g., $H_t^{-1}B_t$ for all $t$ in parallel (similarly with $A$). Following this, we can then compute $A(H^{-1}A^\top)$ in parallel also using the same argument.

In addition, we can solve the block tridiagonal linear systems involving $AH^{-1}A^\top$ by using a specialized parallel solver [5, 36, 27, 41, 15, 18]. Unfortunately, none of these methods have open source code available, and so in our experiments, we implement the simple block tridiagonal matrix algorithm. Implementing a robust parallel solver is an important direction for future work, and will further improve the scalability and numerical stability of IDOC.

**Numerical Stability.** In addition to parallelization, another benefit of avoiding Riccati-style recursions for computing derivatives is improved numerical stability. For long trajectories and/or poorly conditioned COC problems (e.g., stiff dynamics), IDOC reduces the rounding errors that accumulate in recursive approaches. We show superior numerical stability in our experiments compared to PDP in Section 6, despite using the simple recursive block tridiagonal matrix algorithm for solving $AH^{-1}A^\top$. Replacing this recursion with a more sophisticated block tridiagonal solver should further improve the stability of the backwards pass and is left as future work.

## 5.2 Vector Jacobian Products

Another benefit of explicitly writing out the matrix equations for $D\xi(\theta)$ given in Equation 4 is that we can now directly compute VJPs given some outer loss $\mathcal{L}(\xi) \in \mathbb{R}$ over the optimal trajectory. Let $v \triangleq D_\xi \mathcal{L}(\xi)^\top \in \mathbb{R}^{n_\xi \times 1}$ be the gradient of the loss w.r.t. trajectory $\xi(\theta)$. The desired gradient $D_\theta \mathcal{L}(\xi(\theta))$ is then given by $D_\theta \mathcal{L}(\xi) = v^\top D\xi(\theta)$ using the chain rule. The resultant expression is

$$D_\theta \mathcal{L}(\xi(\theta)) = v^\top (H^{-1}A^\top (AH^{-1}A^\top)^{-1}(AH^{-1}B - C) - H^{-1}B). \tag{6}$$

---

[4]Jin et al., [20] claim that the backward pass for DiffMPC is $O(T^2)$ due to direct inversion of the KKT matrix. However, DiffMPC actually solves an auxiliary LQR problem for $O(T)$ complexity.

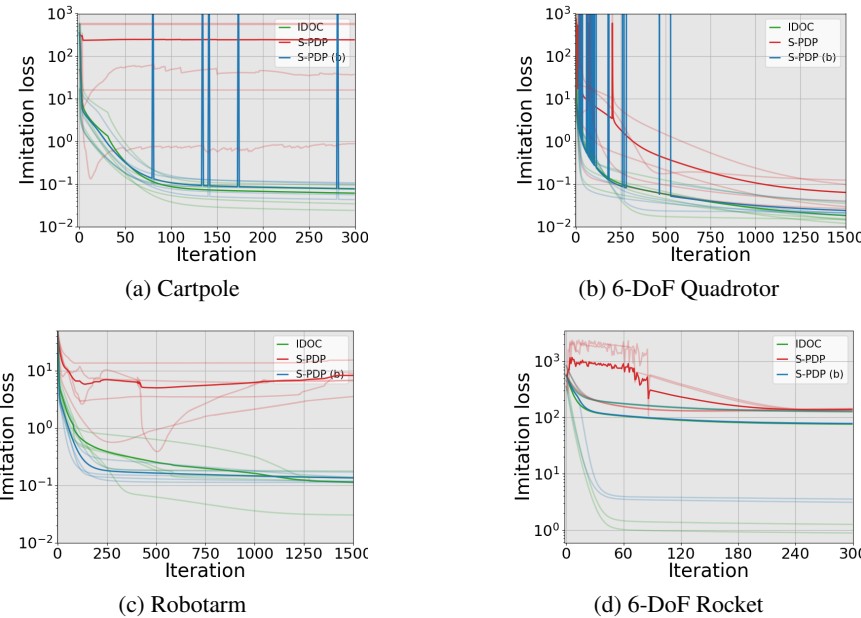

(a) Cartpole

(b) 6-DoF Quadrotor

(c) Robotarm

(d) 6-DoF Rocket

Figure 3: Learning curves for the imitation learning task over five trials. Bold lines represent mean loss, lighter lines represent individual trials. IDOC yields more stable gradients and lower final imitation loss across all environments and trials compared to both Safe-PDP (S-PDP) and Safe-PDP with log-barrier functions (S-PDP (b)).

The simple observation here is that we do not need to construct $D_\theta \xi^\star(\theta)$ explicitly, and can instead evaluate the VJP directly. We propose evaluating Equation 6 from left to right and block-wise, which will reduce computation time compared to explicitly constructing $D\xi(\theta)$ and then multiplying with $v$.

To see this, we follow the example in Gould et al. [14] and assume the blocks in $H$ have been factored. Then for a single block (ignoring constraints for simplicity), evaluating $v^\top (H_t^{-1} B_t)$ is $O((n+m)^2 p)$ while evaluating $(v^\top H_t^{-1}) B_t$ is $O((n+m)^2 + (n+m)p)$. Evaluating the VJP directly significantly reduces computation time compared to constructing the full trajectory derivative for problems with higher numbers of state/control variables and tunable parameters.

## 6 Experiments

While our contributions are largely analytical, we have implemented the identities in Equation 4 to verify our claims around numerical stability and computational efficiency on a number of simulated COC environments. We will show that IDOC is able to compute trajectory derivatives significantly faster compared to its direct alternative PDP [20] and Safe-PDP [22] with superior numerical stability.

### 6.1 Experimental Setup

We evaluate IDOC against PDP [20] and Safe-PDP [22] in an LfD setting across four simulation environments proposed in Jin et al. [22], as well as a synthetic experiment. The simulation environments showcase the gradient quality for a realistic learning task, whereas the synthetic experiment is designed to measure numerical stability and computation times. For Safe-PDP, we evaluate against the method proposed for inequality constrained problems (Safe-PDP), as well as using an approximate log-barrier problem in place of the full problem (Safe-PDP (b)). The IPOPT solver [45] is used in the forward pass to solve the COC problem, and Lagrange multipliers $\lambda$ are extracted from the solver output. More generally however, note that the method proposed in Gould et al. [14] can be used to recover $\lambda$ if another solver is used where $\lambda$ is not provided.

**Imitation Learning/LfD.** The LfD setting involves recovering the model parameters $\theta$ that minimizes the mean-squared imitation error to a set of $N$ demonstration trajectories, given by

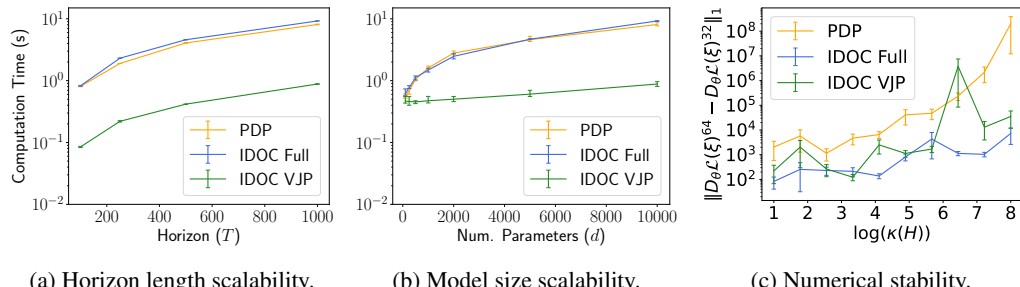

(a) Horizon length scalability.  (b) Model size scalability.  (c) Numerical stability.

Figure 4: Synthetic experiments measuring a) scalability with horizon length $T$, with fixed parameter size $|\theta| = 10k$, b) scalability with number of parameters $d$ with fixed horizon length $T = 1k$, c) numerical stability with varying block condition number $\kappa(H_t)$. Error bars measure standard error across 5 and 25 samples for computation time (a, b) and numerical stability (c), respectively.

$\mathcal{L}(\xi(\theta), \xi^{\text{demo}}) \triangleq \frac{1}{N} \sum_i \|\xi(\theta)_i - \xi_i^{\text{demo}}\|^2$. We include all parameters in the cost, dynamics and constraint functions in $\theta$ and furthermore, evaluate performance both with (S-PDP) and without (PDP) inequality constraints. We perform experiments in four standard simulated environments: cartpole, 6-DoF quadrotor[5], 2-link robot arm and 6-DoF rocket landing. For a detailed description of the imitation learning problem as well as each COC task, see the appendix. In addition, we provide timings for all methods within the simulation environments in the appendix.

**Synthetic Benchmark.** The simulation experiments described above are not sufficiently large-scale to adequately measure the benefits of parallelization and numerical stability afforded by IDOC over PDP. To demonstrate these benefits, we constructed a large-scale, synthetic experiment where the blocks required to construct $H, A, B$ and $C$ (as discussed in Section 4) are generated randomly. Computation time against the horizon length $T$, as well as the number of parameters $d$ is reported. Numerical stability is measured by comparing mean-absolute error between trajectory derivatives evaluated under 32-bit and 64-bit precision for varying condition numbers over the blocks of $H$. State and control dimensions are fixed at $n = 50$ and $m = 10$. Experiments are run on an AMD Ryzen 9 7900X 12-core processor, 128Gb RAM and Ubuntu 20.04. See the appendix for more details.

## 6.2 Imitation Learning/LfD Results

In this section, we evaluate the effectiveness of gradients produced from IDOC against PDP and Safe-PDP [20, 22] for the imitation learning/LfD task. In the equality constrained setting, IDOC and PDP yield identical (up to machine precision) trajectory derivatives and imitation loss throughout learning; see the appendix for more detailed results and analysis. However, as shown in Figure 3, when inequality constraints are introduced, the derivatives produced by Safe-PDP and IDOC differ significantly. Safe-PDP fails to reduce the imitation loss due to unreliable gradients, whereas IDOC successively decreases the imitation loss. While Safe-PDP (b), which differentiates through a log-barrier problem, also provides stable learning, it ultimately yields higher imitation loss compared to IDOC. This is because

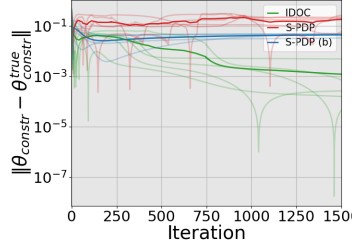

Figure 5: Control Limit Learning.

the barrier problem is an approximation to the true COC problem. Further discussion around using log-barrier methods and COC problems is provided in Section 7.1.

## 6.3 Synthetic Benchmark Results

Figure 4 illustrates the results of the synthetic benchmark. We observed that IDOC and PDP have similar computation time and scalability with problem size when computing full trajectory derivatives. However, computing VJPs with IDOC significantly reduces computation time and improves scalability

---

[5]Interestingly, $H$ is singular in this case, relating to block $H_T$, so we use the trick proposed in Section 4.3 with $\delta = 10^{-6}$ to compute Equation 4. More details around this are provided in the appendix.

by an order of magnitude with model size (i.e., $d = |\theta|$). Numerical stability experiments show that IDOC (full and VJP) yields lower round-off errors and improved stability compared to PDP. Using more sophisticated block tridiagonal solvers will further improve stability.

# 7 Discussion and Future Work

## 7.1 Differentiating through Log-Barrier Methods

In the imitation learning setting, we assume that demonstration data $\xi^{\mathrm{demo}}$ is an optimal solution to a COC problem with (unknown) parameters $\theta^\star$. Furthermore, we assume that a subset of timesteps $\mathcal{A} \subseteq \{1, \ldots, T\}$ yield active constraints. Under the log-barrier formulation, the constraint boundaries must be relaxed beyond their true values during learning to minimize the imitation loss. Concretely, under $\theta^\star$, $\log g_t(x_t^{\mathrm{demo}}, u_t^{\mathrm{demo}}; \theta^\star)$ are undefined for $t \in \mathcal{A}$. Therefore to recover $\xi(\theta) = \xi^{\mathrm{demo}}$, we must have $g_t(x_t(\theta), u_t(\theta)) < 0$ for $t \in \mathcal{A}$, i.e., we cannot recover the true constraint function through minimizing the imitation loss. We verify this using the robot arm environment, and present the results in Figure 5, plotting the error between the estimated and true constraint value. We observe that IDOC recovers the true constraint value more closely compared to Safe-PDP (b).

**Generality of IFT.** The generality of the differentiable optimization framework, and the matrix equation formulation for trajectory derivatives given in Equation 4 yield additional conceptual benefits which may help us tackle even broader classes of COC problems. For example, we can relax the additive cost assumption and add a final cost defined over the full trajectory such as

$$h_T(\xi; \theta) = \xi^\top Q \xi, \tag{7}$$

where $Q = CC^\top$ and $C \in \mathbb{R}^{n_\xi \times k}$ for $k \ll n_\xi$ is full rank. For IDOC, we can simply use the matrix inversion lemma to invert $H$ in $O(T)$. However, this is more complicated for methods derived from the PMP which rely on the specific additive cost structure of the underlying COC problem.

**Dynamic Games.** A generalization of our work is to apply the differentiable optimization framework to handle learning problems that arise in dynamic games [3] and have recently begun to employ PDP-based approaches [10]. Being able to differentiate through solution and equilibrium concepts that arise in dynamic games enables the solution of problems ranging from minimax robust control [3] to inverse dynamic games [10, 33], and is a promising direction for future work.

**Fast forward passes.** While we have proposed a more efficient way of computing trajectory derivatives (i.e., the backward pass), we observe that the forward pass to solve the COC problem bottlenecks the learning process. Significant effort must be placed on developing fast solvers for COC problems for hardware accelerators to allow learning to scale to larger problem sizes.

**Combining MPC and Deep Learning.** A recent application for computing trajectory derivatives is combining ideas from deep learning (including reinforcement learning) with MPC [38, 50, 46, 48]. Common to these approaches is using a learned model to select the parameters for the MPC controller. By allowing for robust differentiation through a broader class COC problems compared to previous approaches, we hope that IDOC will allow for future development in this avenue of research.

# 8 Conclusion

In this paper, we present IDOC, a novel approach to differentiating through COC problems. Trajectory derivatives are evaluated by differentiating KKT conditions and using the implicit function theorem. Contrary to prior works, we do not solve an auxiliary LQR problem to efficiently solve the (differential) KKT system for the trajectory derivative. Instead, we apply variable elimination on the KKT system and solve the resultant matrix equations directly. We show that linear time evaluation is possible by appropriately considering equation structure. In fact, we show that IDOC is faster and more numerically stable in practice compared to methods derived from the PMP. We hope that our discussion connecting the fields of inverse optimal control and differentiable optimization will lead to future work which enables differentiability of a broader class of COC problems.

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

# A Experimental Setup for Imitation Learning/LfD Task

We use the code and simulation environments provided by [22] and follow a similar experimental setup, described in more detail in this section. The LfD experimental setting without inequality constraints is referred to in [20] as the inverse reinforcement learning (IRL) task, whereas including inequality constraints is referred to in [22] as the constrained inverse optimal control (CIOC) task.

**Demonstration Trajectories.** Up to five demonstration trajectories are used in the LfD setting with no inequality constraints. Each demonstration trajectory is generated by solving a COC problem using the same underlying parameters $\theta$, however the initial conditions differ across trajectories and are sampled randomly. For the setting with inequality constraints present, we use only one demonstration trajectory.

**Initialization.** Consistent with [20, 22], we run five imitation learning trials for all LfD experiments, where for each trial, $\theta$ is initialized by adding uniform noise to the true value. For all methods, we use the gradient descent with the same learning rate for a given environment.

**Environment Specifications.** Table 1 provides summary specifications of the simulation environments used in our experiments, which are consistent with prior works [20, 22]. Recall that $n$, $m$ denote the number of states and controls, respectively. The number of parameters is denoted $d = |\theta|$ and $T$ is the horizon length.

Table 1: Description of environments

| Environment | $n$ | $m$ | $d$ | $T_{\text{IRL}}$ | $T_{\text{CIOC}}$ |
|---|---|---|---|---|---|
| Cartpole | 4 | 1 | 9 | 30 | 35 |
| Quadrotor | 13 | 4 | 11 | 50 | 25 |
| Robotarm | 4 | 2 | 10 | 35 | 25 |
| Rocket | 13 | 3 | 12 | 40 | 40 |

**Additional Environment Parameters.** Additional specifications are presented in Table 2. These include log-barrier parameter $\gamma$ used for Safe-PDP (b), time discretization for Forward Euler to discretize continuous time dynamics, the learning rate for minimizing the imitation loss and finally the number of demonstration trajectories per environment. (E) refers to experiments without constraints, whereas (I) means inequality constraints are present.

Table 2: Additional hyperparameters for LfD experiments

| Environment | $\gamma$ | $\Delta$ | lr | $n_{\text{demos}}$ |
|---|---|---|---|---|
| Cartpole (E) | - | 0.1 | $10^{-4}$ | 5 |
| Quadrotor (E) | - | 0.1 | $10^{-4}$ | 2 |
| Robotarm (E) | - | 0.1 | $10^{-4}$ | 4 |
| Rocket (E) | - | 0.1 | $3 \times 10^{-4}$ | 1 |
| Cartpole (I) | 0.01 | 0.1 | $8 \times 10^{-5}$ | 1 |
| Quadrotor (I) | 0.01 | 0.15 | $2 \times 10^{-4}$ | 1 |
| Robotarm (I) | 0.01 | 0.2 | $2 \times 10^{-3}$ | 1 |
| Rocket (I) | 1 | 0.1 | $10^{-5}$ | 1 |

## A.1 Cartpole

The cartpole environment relates to the swing-up and balance task, where a simple pendulum (i.e., massless pole and point mass on the end of the pole) is swinging on a cart and the objective is to balance the pendulum above the cart. Only a horizontal force can be applied to drive the cart.

**Dynamics.** The state variable $x = [y, q, \dot{y}, \dot{q}]^\top \in \mathbb{R}^4$ is comprised of the horizontal position of the cart $y$, counter-clockwise angle of the pendulum $\theta$ (from the hanging position) and their respective velocities. Control $u \in \mathbb{R}$ relates to the horizontal force applied to the cart. Dynamics parameters $\theta_{\text{dyn}} = [m_c, m_l p, \ell]^\top \in \mathbb{R}^3$ relate to the the mass of the cart $m_c$, mass of the point mass at the end of the pole $m_p$ and length of the pole $\ell$. Gravity is set at $g = 10$. The continuous time dynamics are given by

$$\ddot{y} = \frac{1}{m_c + m_c \sin^2 q} \left[ u + m_p \sin q (\ell \dot{q}^2 + g \cos q) \right] \tag{8}$$

$$\ddot{q} = \frac{1}{\ell(m_c + m_p \sin^2 q)} \left[ -u \cos q - m_p \ell \dot{q}^2 \cos q \sin q - (m_c + m_p) g \sin q \right], \tag{9}$$

with a detailed derivation provided in [43, Ch. 3].

**Cost Functions.** The cost function is a quadratic cost over states and controls. Specifically,

$$c_t(x_t, u_t; \theta) = (x_t - x_{\text{goal}})^\top W (x_t - x_{\text{goal}}) + w_u u^2, \tag{10}$$

and

$$c_T(x_T, u_T; \theta) = (x_T - x_{\text{goal}})^\top W (x_T - x_{\text{goal}}), \tag{11}$$

with $x_{\text{goal}} = [0, 0, -\pi, 0]^\top$. We specify that $W \triangleq \text{diag}(w)$, where $w = [w_y, w_q, w_{\dot{y}}, w_{\dot{q}}] \geq 0$ and furthermore, $w_u \geq 0$. We have that $\theta_{\text{cost}} = [w^\top, w_u^\top]^\top \in \mathbb{R}^5$.

**Constraints.** For the setting with constraints, we apply box constraints for the cart position and controls. Specifically, we enforce $|y| \leq y_{\text{max}}$ and $|u| \leq u_{\text{max}}$. We let constraint parameters $\theta_{\text{constr}} = [y_{\text{max}}, u_{\text{max}}]^\top$. The final parameter vector is given by $\theta = [\theta_{\text{dyn}}^\top, \theta_{\text{cost}}^\top, \theta_{\text{constr}}^\top]^\top$.

## A.2  6-DoF Quadrotor Manouvering

See Section E of the appendix in [20] for the full definition of the quadrotor maneuvering problem. The objective is to drive the quadrotor using propeller thrusts to a goal configuration. The state is given by $[p^\top, \dot{p}^\top, q^\top, \omega^\top]^\top \in \mathbb{R}^{13}$, where $p \in \mathbb{R}^3$ represents the position of the center-of-mass of the quadrotor, $\dot{p} \in \mathbb{R}^3$ represents linear velocity, $q \in \mathbb{R}^4$ is a unit quaternion representation of the quadrotor attitude and $\omega \in \mathbb{R}^3$ represents angular velocity. For the inequality constrained setting, non-linear (squared-norm) constraints on quadrotor position, given by $\|p\|_2^2 \leq r_{\text{max}}$ and thrust limits are applied.

## A.3  2-Link Robot Arm

See Section E in the appendix in [20] for a reference to the full description of the robot arm problem. The objective of this OC problem is to move the robot arm to a goal configuration. The state is given by the orientation of the base and elbow joint, as well as respective velocities $[q_1, q_2, \dot{q}_1, \dot{q}_2]^\top \in \mathbb{R}^4$. The control inputs $u = [u_q, u_2]^\top \in \mathbb{R}^2$ relate to applying torques at each joint. Constraints for the inequality constrained setting relate to box constraints on the joint positions, as well as torque limits. The cost function is similar to cartpole, i.e., a quadratic penalty to a goal state and over the controls.

## A.4  6-DoF Rocket Landing

See Section I in the appendix in [20] for the full definition of the 6-DoF rocket landing problem. The objective of this task is to land a rocket modeled as a rigid body (softly) onto a goal position, using thrusters located at the tail. A non-linear constraint over the tilt angle is applied in the inequality constrained setting. The cost function penalizes deviations from the goal configuration, as well as fuel cost.

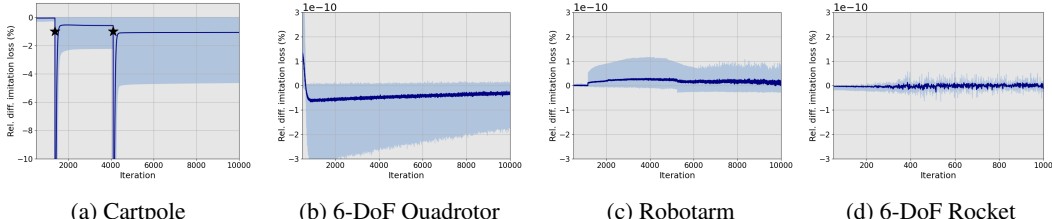

|  (a) Cartpole | (b) 6-DoF Quadrotor | (c) Robotarm | (d) 6-DoF Rocket |

Figure 6: Results for the imitation learning task with no inequality constraints present. The y-axis is the (percentage) relative difference in the imitation loss, defined to be (IDOC loss - PDP loss) / PDP loss (lower is better). For Cartpole, locations where the PDP gradient fails is marked by stars. Shaded regions indicate min/max values over five trials.

## B   Experimental Setup for Synthetic Benchmark

We discuss in more detail how blocks for $H, A, B, C$ are generated in this section. Elements for the Jacobian and Hessian blocks for the objective function and constraints are sampled independently and identically distributed (i.i.d.) from a standard Gaussian. Hessian blocks are made symmetric using $H^{\text{sym}} = (H + H^\top)/2$, where $H$ is the initially generated random matrix. Each block's condition number is modified by applying an SVD to each block and adjusting the diagonal entries. Elements of the downstream loss gradients $v = D_\xi \mathcal{L}(\xi)^\top$ are also sampled i.i.d. from a standard Gaussian.

## C   Additional Results for the Imitation Learning/LfD Tasks

In this section, we provide additional detailed analysis around the imitation learning/LfD setting with no inequality constraints. Figure 6 illustrates the results of these experiments. We plot the percentage difference in loss across the full training curve between IDOC and PDP for five random initializations for $\theta$. For the quadrotor, robotarm and rocket environments, we observe almost no difference in learning curves. This is expected since IDOC and PDP are computing the same trajectory derivative. For cartpole however, we see that PDP fails catastrophically (resulting in undesirable spikes in the imitation loss) on two occasions across the five trials. We suspect this is due to numerical instability of the Riccati equations, arising from the local geometric properties of the dynamics at solved trajectories.

## D   Additional Timings for Simulation Experiments

In this section, we provide additional compute time experiments for the imitation learning/LfD tasks evaluated in the simulation environments. Given the relatively small size for the COC problems, there is very limited advantage with respect to compute time for parallel evaluation of trajectory derivatives. However, as a proxy to parallelization, IDOC can be *vectorized* across timesteps using the numerical linear algebra library Numpy [16]. All experiments in this section are run on a single thread of an AMD Ryzen 9 7900X 4.7Ghz desktop CPU.

The experimental setting where inequality constraints are present is challenging to vectorize compared to the setting with only equality constraints, because different timesteps may have different numbers of active inequality constraints. While matrices $A$ and $C$ presented in Equation 4 are still block structured, the blocks may not be of uniform size in this setting. We present in Figure 7a and 7b the mean computation time (along with the upper/lower bound) per iteration across five trials with 10k iterations for each trial for all environments except for rocket (1k). We observe around a $2\times$ speedup over PDP by computing our IFT/DDN gradients, even before direct computation of vector-Jacobian products. Computing VJPs directly yields a further (approx.) 10% saving in compute time over constructing the trajectory derivative explicitly.

In our non-optimized Python implementation of IDOC, we batch together and vectorize computations involving blocks with an identical number of active constraints. As expected, trajectory derivatives for inequality constrained problems are slightly slower to compute compared to the equivalent without (hard) inequality constraints (log-barrier approximation), due to the necessary computational overheads required for identifying and batching blocks.

Note that Safe-PDP [22] (for hard inequality constraints) and PDP [20] (used for differentiating through log-barrier problems) derivatives are implemented using a custom LQR and equality constrained LQR solver, respectively. These solvers have markedly different implementations, which explains differences in computation time between Safe-PDP and PDP.

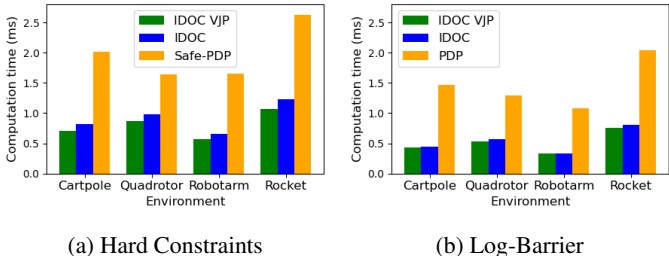

(a) Hard Constraints  (b) Log-Barrier

Figure 7: Computation times for trajectory derivatives (per trajectory, per iteration) for the CIOC task. As expected, the log-barrier method with no "hard" inequality constraints affords faster derivative computation.

