# OpenReview forum: "Revisiting Implicit Differentiation for Learning Problems in Optimal Control"
_NeurIPS.cc/2023/Conference — NeurIPS 2023 poster_

### Official Review · Reviewer_dzG6 · 2023-07-02

**Soundness:** 3 good
**Presentation:** 3 good
**Contribution:** 3 good
**Rating:** 6
**Confidence:** 3

**Summary:**

This paper considers the optimization problem in the context of discrete-time control of dynamic systems. The paper proposes a method for efficiently and effectively evaluating the Jacobian of saddle points of constrained optimal control problems w.r.t. COC problem parameters. The main technical result is that the computation cost grows linearly with the number of time steps. The proposed approach is evaluated on standard environments like quadrotors and cart-poles.

**Strengths:**

The whole paper is well-written and easy to follow. I appreciate the writing in the technical section (Secs. 3-5), which unfolds the core technical method in a nice and coherent way.

The main technical idea is solid and sound. Fig. 2 turned out to be quite helpful when I was checking the equations. The linear time complexity follows naturally after Proposition 1 is established.

**Weaknesses:**

I don’t see major fallacies in the main technical result itself, but I still have a few high-level concerns regarding the novelty and usefulness of the proposed technique:

- I feel that Propositions 1, 2, and Fig 2 are direct results of applying existing numerical and mathematical techniques in derivative derivations. From what I understand, no new tools were developed in proving these results. As I am not well-calibrated with NeurIPS’s bar, I will let other reviewers decide how novel the technical method is.

- The writing in the abstract and introduction gives me the impression that the gradient computation is fully parallelizable. I can see how multiplications involving H^(-1) can be parallelized. However, solving the tridiagonal block matrix seems like a sequential procedure to me. I am guessing from reading Lines 234-239 that the implementation used a sequential solver with O(T) time complexity and was the major time bottleneck (correct me if I am wrong). It would be useful to profile the time cost of each step that can and cannot be parallelized.

- The experiments didn’t seem to evaluate the proposed method in large-scale problems. Dynamical systems (and their gradients) like quadrotor can be computed extremely fast (orders of magnitude than real time) because of their few degrees of freedom. The number of time steps (<=50 if I understand correctly from Table 1) does not seem to create a large-sized Hessian (a few hundred by a few hundred?) challenging enough to compute for modern CPUs. Having 2x speed up is still nice, but I had a higher expectation for a parallel (?) linear algorithm given that the baseline is sequential and quadratic.

- On a related note, I didn’t find visualizations of the quadrator/cart-pole/etc environments in the main paper or the supplemental material. Having some images to visualize these tasks would be great.

A very minor comment on writing: Some sentences are quite long and hard to follow, e.g., lines 21-23 and lines 31-34. Splitting them into several small sentences would be better.

Some technical comments:

- Lines 124 and 125: is R^p a typo? It makes more sense to me if we replace R^p with R^d as d is the dimension of \theta. What is p?
- Line 150: Why is a transpose operator needed?
- Proposition 1: same question about p in the definitions of B and C; should it be replaced with d?
- Proposition 1: Why is C defined as the Jacobian of h w.r.t. \theta? It makes more sense to me if it is the Jacobian of r w.r.t. \theta, which is also implied by the dimension n_r.
- Proposition 2: I feel the linear time complexity O(T) assumes the dimension of \theta does not grow with the number of time steps, or H^(-1)B won’t be computable in O(T) time. Please correct me if I am wrong.
- Eqn. (5): it seems more common to flip the sign of lambda and replace -A^T with A^T so that the matrix is symmetric.

**Questions:**

See my comments above.

**Limitations:**

See my comments above.

---

> ### Author Rebuttal · Authors · 2023-08-09
>
> Thank you for your detailed review and kind words around the technical writing.
>
> > **From what I understand, no new tools were developed in proving these results.**
>
> We agree that no new mathematical tools were developed. However, identifying and exploiting the link between the IFT identities of Gould et al. and optimal control is non-trivial, as evidenced by it being missed by both the learning and control communities. We specifically want to debunk prior statements we observed in the literature that approaches based on IFT identities are inherently quadratic.
>
> We believe that such assertions need to be refuted formally by demonstrating that there are surprisingly meaningful numerical scalability and stability improvements offered by the novel application of IFT identities to control. We believe that our results (both theoretical and numerical) therefore represent an important novel contribution to both the learning and control communities.
>
> > **However, solving the tridiagonal block matrix seems like a sequential procedure to me. I am guessing from reading Lines 234-239 that the implementation used a sequential solver with O(T) time complexity and was the major time bottleneck (correct me if I am wrong).**
>
> Thank you for drawing our attention to the apparent disconnect between our discussion of parallelization and the solution of the tridiagonal block matrix. We will clarify our discussion in the abstract and introduction by highlighting that it is currently easy to parallelize everything except the solution of the tridiagonal block matrix, but that there are a growing number of methods (some of which we cite) that propose parallelizable block tridiagonal solvers.
>
> Unfortunately, the code for these parallelizable block tridiagonal solvers appears unavailable. As a result, we implement a sequential procedure for the block tridiagonal system in Python. Despite this, we still observe significant improvements in computational performance. We envision that future availability and development of custom (parallelizable) block tridiagonal solvers will yield further reductions in computation time.
>
>
> > **It would be useful to profile the time cost of each step that can and cannot be parallelized.**
>
> For $p=10$k and $T=1$k, our most challenging setting, we observe that the serial computation (block tridiagonal solve using the Thomas algorithm) occupies approx. 45% of total compute time for IDOC full, whereas for IDOC VJP it occupies 18%. For a fixed $p$, this ratio does not change with horizon length $T$.
>
> The ratio for $p=1$k is 39% and 36% for IDOC full and VJP, respectively. This indicates that the importance of the serial computation diminishes as the problem size (w.r.t. parameters) increases. The serial component is still significant however, and warrants developing parallel block tridiagonal solvers to further improve scalability.
>
> > **The experiments didn’t seem to evaluate the proposed method in large-scale problems.**
>
> We appreciate you pointing out that our first set of experiments, whilst showing a 2x speedup, could have been more compelling. We have therefore conducted additional large-scale experiments as detailed in the global response and attached pdf in which the dimensions of the parameters, states, and controls are large enough for parallelization to yield benefits over serial computation.
>
> These additional results (cf. Fig. 1) offer compelling evidence that our IDOC VJP method is approximately 10x faster across all horizon lengths $T$ for a large-scale problem ($p=10000$), with further gains possible as the number of learnable parameters increases.
>
> > **On a related note, I didn’t find visualizations of the quadrator/cart-pole/etc environments in the main paper or the supplemental material. Having some images to visualize these tasks would be great.**
>
> We will add visualisations (or suitable references) for the benchmark problems to the supplemental material.
>
> > **A very minor comment on writing: Some sentences are quite long and hard to follow, e.g., lines 21-23 and lines 31-34. Splitting them into several small sentences would be better.**
>
> We agree that some sentences in the submission are too long and could be made clear, thank you for bringing this to our attention. Particular attention will be paid to those you have identified.
>
> ### Technical Comments
>
> Thank you for carefully reading our work and picking up these typos! We will address these in order.
>
>
> * Yes, this is a typo! $p$ should be $d$.
> * We assume $\text{D}\mathcal{L}_\xi(\xi)\in \mathbb{R}^{(n+m)T + n \times 1}$ is a column vector. We will make this more explicit.
> * Correct! $p$ should be $d$ here as well.
> * Yes, this should indeed be $r$.
> * As is common in (optimal) control, we assume that $\theta$ is static across time and is comprised of all relevant parameters across all (time-varying) cost and constraint functions.
> * Agreed, happy to make it symmetric and perform a sign change elsewhere.

---

> > ### Comment · Reviewer_dzG6 · 2023-08-20
> > **Thank you for the response**
> >
> > After reading the rebuttal, I remain supportive of the paper and will raise my score to weak accept. I think there is still room for improving the experiments, but I am OK with the current paper with all the promised changes incorporated.

---

### Official Review · Reviewer_fBmg · 2023-07-04

**Soundness:** 3 good
**Presentation:** 4 excellent
**Contribution:** 3 good
**Rating:** 6
**Confidence:** 5

**Summary:**

This paper analyzes and develops an efficient method to differentiate through a constrained discrete-time optimal control system, i.e., computing the derivative of an optimal trajectory of a constrained discrete-time optimal control system with respect to the parameters in the system’s cost function, dynamics, and/or constraints, based on the implicit function theorem (IFT). This is a key problem in many learning problems, such as inverse optimal control (inverse reinforcement learning, system identification, or learning from demonstration). The contributions of this paper include (1) the authors analyzed the sparse structures of IFT equations, and showed that the linear time complexity (in the trajectory horizon) can be obtained for solving IFT, instead of quadratic complexity. (2) Based on the analysis, the authors develop an efficient algorithm by parallelization and a vector jacobian auto-diff algorithm. The algorithm is evaluated in comparison the state-of-the-art (PDP [18] and Safe-PDP [20]) for various tasks,  the proposed method is shown at least 2x faster and more numerical stability.


**Strengths:**

The study of differentiable optimal control is well-motivated, and the problem relates to many problems in learning and robotics.
The authors analyzed sparse structure of the IFT equation for the constrained optimal control problem, and showed the linear complexity (in trajectory horizon) of differentiating through optimal control. Besides, they established the connection between solving IFT equation and PDP. Based on the analysis, they develop parallelize and vector Jacobian Products algorithms to accelerate the backward pass, which have been shown effective and stable.

The algorithm is novel and has shown obvious improvement over the existing method (mainly PDP and Safe-PDP). I believe the algorithm will be useful and of interest of control and learning communities. The paper is well organized and well presented, and I expect the proposed method is important in practice,


**Weaknesses:**

In order to further improve the paper, some claims may need to be further clarified.

- Line 159: since the algorithm requires the identification of a set of active constraints, \tilde{g}_t, from all inequality constraints, will the use of a threshold cause the numerical issues, eventually leading to bad quality gradient and unstable gradient descent?
- In Fig. 4 cartpole example, is the instability of Safe PDP without log-barrier functions because of the numerical issues of identifying active and inactive inequality constraints?
- In my understanding, Safe PDP using log-barrier function has two benefits: first, it avoids the needs of active identification of inequality constraints, and 2) it creates some smoothing effects in the trajectory solution space because the non-differentiability caused by the switch between “active” and “inactive” constraints in the change of parameters. However, as the authors point out, Safe PDP with barrier function is only an approximation method to compute the gradient, thus it is not surprising to have a larger loss, as shown in Fig.4.
- Currently, most of the examples in experiments consider the loss and stability performance.  In order to support the claim of "2x" more efficiency, I think more experiments about algorithm speed test (with respect to PDP) should be done.





**Questions:**

Please find my question in the "weaknesses" section.

**Limitations:**

Please find in the "weaknesses" section.

---

> ### Author Rebuttal · Authors · 2023-08-09
>
> Thank you for your review. We will address your questions as follows:
>
> > **Line 159: since the algorithm requires the identification of a set of active constraints, \tilde{g}_t, from all inequality constraints, will the use of a threshold cause the numerical issues, eventually leading to bad quality gradient and unstable gradient descent?**
>
> This is potentially an issue in practice, and this instability is well known when designing numerical solvers which use the so-called ``active-set" approach. While we didn't experience issues around constraint switching in our experiments, we acknowledge that it could be an issue in practice and may be encountered when scaling these approaches to more difficult problems.
>
> > **In Fig. 4 cartpole example, is the instability of Safe PDP without log-barrier functions because of the numerical issues of identifying active and inactive inequality constraints?**
>
> We're not sure if the instability of Safe-PDP without barrier functions is caused by identification of active and inactive constraints. While this claim is made in the Safe PDP paper (bottom on page 9 under Problem III), our approach IDOC also needs to identify the active constraints. Of course, we use the same threshold for identifying constraints for a fair comparison. We believe that constraint switching and identification in and of itself is not the root cause of instability for Safe-PDP. We will add this commentary into the manuscript since this is a very useful piece of analysis which arises from the experimental results.
>
> > **In my understanding, Safe PDP using log-barrier function has two benefits: first, it avoids the needs of active identification of inequality constraints**
>
> We believe that the identification of active/inactive constraints in and of itself is a fast, simple operation (evaluate constraint functions and apply threshold). However, varying numbers of active constraints along the trajectory make it difficult to vectorize the computation of the trajectory derivatives, due to varying-sized blocks.
>
> This property generally increases computation time compared to using the log-barrier method. Otherwise, the identification of constraints does not appear (at least, in our experiments) to impact gradient quality.
>
> >**it creates some smoothing effects in the trajectory solution space because the non-differentiability caused by the switch between “active” and “inactive” constraints in the change of parameters.**
>
> We agree that the smooth log-barrier function may yield superior gradient information during learning, since constraints don't need to be satisfied to provide gradient information. This may explain why the log-barrier approach reduces imitation loss faster at the start compared to IDOC.
>
> However, as you correctly point out, IDOC provides lower final imitation loss for many of the experiments. This appears to be due to IDOC correctly identifying the hard constraint values. We provide further insight around this in the global response.

---

### Official Review · Reviewer_tsv7 · 2023-07-04

**Soundness:** 3 good
**Presentation:** 2 fair
**Contribution:** 3 good
**Rating:** 5
**Confidence:** 4

**Summary:**

There has been recent interest in differentiating through trajectories to obtain first-order derivatives for optimization problems including policy learning, inverse optimal control and model learning. Previous work usually uses a backward recursion which computation scales quadratically with the length of the horizon. In this work, the derivatives are computed exploiting the block-diagonal structure with computational time linearly varying with the length of the horizon. Additionally, it provides the flexibility of parallelizing the computation. This method can also directly compute vector-jacobian products easily. Most of these advantages are verified through training iterations with imitation learning and constrained inverse optimal control tasks on various robotics benchmarks.

**Strengths:**

A novel, interesting method to parallelize and accelerate the computation of derivatives is presented by exploiting the block diagonal structure of matrices. This method is numerically better-conditioned as compared to existing algorithms such as PDP and safe-PDP. Further, it is easily amenable for computing vector-jacobian products which provides it with an advantage.

**Weaknesses:**

The paper is making confusing claims or is making a misclaim.

Line 38 from submission - “Naively applying these identities leads to a quadratic complexity with the length of the trajectory, which is described in prior works [18, 20]” - the citations are PDP and safe PDP.

Line 46 from submission - “Furthermore, we show that the computation of these derivatives is linear with trajectory length, contradicting claims in prior works” - as far as I can tell, the previous backward and forward recursion also scales linearly with trajectory length. The PDP and safe PDP do not explicitly state quadratic complexity with trajectory length anywhere.

Excerpt from PDP: “Third, in the backward pass, unlike differentiable MPC which costs at least a complexity of $O((m+2n)^2 T^2)$ to differentiate a LQR approximation, PDP explicitly solves for first derivative by an auxiliary control system, where thanks to the recursion structure, the memory and computation complexity of PDP is only $O((m+2n)T)$.”

Excerpt from this submission Line 82 - “While the derivative computation in PDP is linear with the number of timesteps, it is inherently a serial calculation, requiring a recursion through time”

In conclusion, the paper is making contradicting claims about advantage over previous work. Otherwise, the paper is not communicating the claims correctly.

I believe whatever computational benefit is reported is because of the ability to parallelize the computation and the final numerical results are above average but not excellent. The numerical stability problem with PDP is not consistently seen and it is likely that it can be quelled by better numerical conditioning.

Additive cost functions with quadratic structure is only one type of optimal control problem. The block diagonal structure is dependent on this assumption. There is a discussion in future work about extensions to non-additive cost functions.



**Questions:**

1) While citing results from Gould et. al such as equation (5) and Proposition 1, the authors are encouraged to provide more explanations about whether the results are a direct application and provide more context in the paper.

2) It is not communicated clearly what is the final take away from sections 4.3 and 4.4. It looks like by assuming H is non-singular, it is possible to do the computation faster using this method. Will this method also result in numerical issues if H is improperly conditioned? If adding the proximal term is the solution, why does a similar type of solution not work for PDP? In other words, add a numerical conditioning to derivatives of c?

3) This paper needs to be self-contained and there is too much dependence on referring to previous papers overall.


**Limitations:**

The authors have discussed the limitations in future work.

---

> ### Author Rebuttal · Authors · 2023-08-09
>
> We appreciate you taking the time to review our paper and drawing our attention to specific claims within it that appeared unclear.
> In particular, we apologise for the confusion regarding our comments around quadratic complexity -- we were not intending to make a misclaim here.
>
> > **The paper is making confusing claims or is making a misclaim.**
>
> To clarify, our comments around quadratic complexity do not relate to the PDP and Safe-PDP methods themselves. They relate to comments in the PDP and Safe-PDP papers about the quadratic complexity of differentiable MPC and CasADi. Reading our submission again, we recognize that this was unclear and apologize for the confusion. The comments that we are referring to are:
>
> * Section 7 in the PDP paper (NeurIPS proceedings version):  *"unlike differentiable MPC which costs at least a complexity of $O((m+2n)^2 T^2)$ to differentiate a LQR approximation"*
> * Section 8 of the Safe-PDP paper (NeurIPS proceedings version): *"Specifically, Safe PDP has a complexity of $O(T)$, while CasADi and Differentiable MPC have at least $O(T^2)$. This is because both CasADi and differentiable MPC are based on the implicit function theorem [75] and need to compute the inverse of a Hessian matrix of the size proportional to $T \times T$.".*
>
> The second comment from the Safe-PDP paper refers to methods using the implicit function theorem (IFT) requiring a matrix inversion of the differential KKT system which is $O(T^3)$ in general. We are showing in our contribution that we can use IFT identities (derived from the same differential KKT system with variable elimination) presented in Gould et al., and recover $O(T)$ complexity. This time complexity does not sacrifice expressiveness; we differentiate through the exact same class of optimal control problems described in the PDP and Safe-PDP paper.
>
> We will include the specific quotes from the PDP/Safe-PDP paper to clarify our actual claims when we update the manuscript. We are simply refuting that methods based on the IFT require an $O(T^3)$ matrix inversion. In fact, we show that using the IFT yields significant benefits to computation time and numerical stability, especially for problems with inequality constraints.
>
> > **This paper needs to be self-contained and there is too much dependence on referring to previous papers overall.**
>
> We will improve the self-sufficiency of this paper as described in the global response. We will also better clarify that equation (5) and proposition 1 are taken directly from Gould et al.
>
> > **I believe whatever computational benefit is reported is because of the ability to parallelize the computation and the final numerical results are above average but not excellent.**
>
> Please see our global response on the additional experiments. We hope this improves our results beyond above average!
>
> > **The numerical stability problem with PDP is not consistently seen and it is likely that it can be quelled by better numerical conditioning.**
>
> We agree that techniques can be used to improve the stability of PDP, for instance using robust matrix factorizations (e.g., SVD instead of Cholesky/LU). However, we believe that alone cannot significantly improve the performance of Safe-PDP for hard constraints. In fact, in the Safe-PDP implementation, SVD is used for matrix inversion/factorization, whereas we simply use an LU-based solver. We believe our experiments show conclusively that the difference between Safe-PDP and IDOC is significant where hard inequality constraints are present, and doubly so given the simple nature of the optimal control problems we evaluate on.
>
> To further reinforce our claims, in the global response and attached pdf, we provide a brief discussion on the importance of using hard constraints instead of log-barrier functions.
>
> > **Additive cost functions with quadratic structure is only one type of optimal control problem. The block diagonal structure is dependent on this assumption. There is a discussion in future work about extensions to non-additive cost functions.**
>
> Additive cost is not the most general formulation, but encompasses many optimal control problems and enables the use of Bellman's principle of optimality. We also point out that PDP and Safe-PDP will not work without an additive cost structure. Recall that our formulation can also extend to include a final cost defined over the whole trajectory, which is not easily achieved under PDP/Safe-PDP.
>
> We also note that a quadratic cost structure is not required to use our proposed method, only that the cost function is twice differentiable.
>
> > **It looks like by assuming H is non-singular, it is possible to do the computation faster using this method. Will this method also result in numerical issues if H is improperly conditioned?**
>
> Your reasoning is correct, non-singularity of $H$ will allow faster and more stable computation. We accept that poor conditioning in $H$ will cause issues for our method, but these issues will also be problematic for PDP and Safe-PDP since they use (and invert) the same blocks used to build $H$ in their calculations.
>
> > **If adding the proximal term is the solution, why does a similar type of solution not work for PDP? In other words, add a numerical conditioning to derivatives of c?**
>
> A small proximal term is important for the quadrotor experiment, where $H$ is inherently singular. The other experiments on the other hand, avoid using the proximal term altogether. Therefore, the numerical differences cannot be attributed to the addition of the proximal term alone.
>
> We hope by addressing your major concerns around the claims, you will consider raising your score.

---

> > ### Comment · Reviewer_tsv7 · 2023-08-13
> > **Acknowledgement**
> >
> > I have read the author response and other reviews. The authors have given a fair response to my questions. The contribution of the paper is better appreciated after this. Many of the differentiable trajectory optimization methods do run into numerical issues. However, I believe the paper can still be improved upon and presented better. Further, it helps to include a further discussion on ILQR/DDP literature from robotics and control. e.g. [1]. At this time, I increase my score to 5.
> >
> > [1] Roulet, Vincent, et al. "Iterative linear quadratic optimization for nonlinear control: Differentiable programming algorithmic templates." arXiv preprint arXiv:2207.06362 (2022).

---

> > > ### Author Response · Authors · 2023-08-16
> > > **Thank you**
> > >
> > > Thank you for acknowledging our response and for raising your score accordingly. We will action feedback from the reviewers to improve the presentation of the paper.
> > >
> > > We are happy to include more discussion around shooting methods used in robotics and control like DDP (and iLQR). We would like to emphasize that the derivation our method (and any method that differentiates through optimality conditions) **does not depend on the solver used to solve the optimal control problem**, as long as the solver successfully finds an optimal point. It is therefore imperative that a robust solver with sufficient convergence guarantees is used to ensure that the trajectories returned by the solver are in fact, optimal.
> > >
> > > For this reason, we picked IPOPT to solve our control problems (consistent with PDP/Safe-PDP). This is due to IPOPT's primal-dual line search filter approach w/second order corrections, which has global convergence properties. As far as we are aware, DDP style solvers have not quite reached the same robustness. We will include this information in the updated version of the manuscript.

---

> > > > ### Comment · Reviewer_tsv7 · 2023-08-16
> > > >
> > > > Good to know this. Please include this information as mentioned. DDP/ILQR also has a backward pass that is linear in trajectory length. Additionally, they do provide closed-loop correction gain matrices which are useful sometimes.

---

### Official Review · Reviewer_CbeP · 2023-07-10

**Soundness:** 2 fair
**Presentation:** 2 fair
**Contribution:** 3 good
**Rating:** 4
**Confidence:** 2

**Summary:**

Prior work shows that computing trajectory derivatives scales quadratically w.r.t to timesteps. This work proposes that trajectory derivatives scales linearly w.r.t. to timesteps, which can be parallelized, resulting in decreased computation time and increased numerical stability.

**Strengths:**

- Good paper structure, reasonable flow.
- Evaluation shows that the gradient computation is numerically stable, which leads to stable training for inverse RL and imitation learning tasks.

**Weaknesses:**

- Section 3.1 is not needed; readers are familiar with matrix and vector derivatives.
- Good to move derivative w.r.t to trajectory in Section 4 as a motivation for this work. (i.e. move to Section 3)
- Need to better structure Section 4; state in beginning of 4.2 the goal is prove that  computing the trajectory derivative relies on a block matrixes.
- Assumes theta is a vector, does not apply for non-linear approximations such as DNNs
- The main benefit of this work is being able to parallelize the computation of trajectory derivatives. There seems to be a lack of evaluation on this. (outside of 1(b)). Need to show total training time.

**Questions:**

-How does this method scale with longer and longer trajectories? Need to see X axis as trajectory length and Y axis as solving time.
- Confused, why prior work fail to identify the block structure? It seems to directly follow from prior work (Gould et al.)

**Limitations:**

Yes

---

> ### Author Rebuttal · Authors · 2023-08-09
>
> Thank you for your review. We would like to emphasize not only the decreased computation time, but the significant performance gains on differentiating through inequality constrained problems without resorting to a log-barrier approximation. Additional compute and scalability experiments are presented in the global response. We will make your suggested changes to the document structure to enhance readability.
>
> > **Assumes theta is a vector, does not apply for non-linear approximations such as DNNs**
>
> While we assume $\theta$ is a vector, this does not restrict its generality. For the DNN case you mention, $\theta$ would represent the parameters of a neural network "unrolled" into a vector. Importantly, we can define expressions such as $df/d\theta$ even for a DNN, which can then be used to compute the expressions in Eq. 4 in the original manuscript. Does this address your concern?
>
> > **Need to show total training time**
>
> The only difference between IDOC and PDP/Safe-PDP is the computation of the backward pass. The total training time is heavily dependent on the solver used in the forward pass (e.g., IPOPT, DDP/iLQR solver, MPC PyTorch solver that runs on GPU). We believe our current comparison between methods is fair (with a very similar implementation in Python/Numpy across methods), and do not wish to muddy the comparison by specifying a full learning system. However, to make our comparisons even more comprehensive, we have provided additional compute and scalability experiments in the global response.
>
> > **Confused, why prior work fail to identify the block structure? It seems to directly follow from prior work (Gould et al.)**
>
> We're not sure how to answer this question, but we believe we are the first to leverage the Gould et al. identities. Admittedly, there is a bit of confusion when reflecting on prior works. Differentiable MPC [2] did correctly identify the block sparse structure in the KKT system and formulated an auxiliary LQR problem to solve it, as reviewer kJe4 correctly pointed out. The Gould et al. identities are different from solving the KKT system as in [2] however, since they are derived by eliminating the Lagrange multipliers (see Section 4.3 in our original manuscript).
>
> This auxiliary LQR approach from [2] is very similar to the approach proposed in PDP and Safe-PDP. However, Safe-PDP claimed that [2] (which uses the implicit function theorem) yields a trajectory derivative that scales quadratically with horizon $T$, which was an important motivation for our work. We will more clearly clarify the difference in approach of our method compared to existing work (e.g., PDP/Safe-PDP/DiffMPC) in the revised manuscript.
>
> Rather than simply identifying the block structure from the identities (recently) derived in Gould et al., we believe exploiting the structure of the resulting equations to demonstrate faster computation and substantially more stable trajectory derivatives across a range of problems (especially through inequality constrained problems) is a significant contribution.
>
> > **-How does this method scale with longer and longer trajectories? Need to see X axis as trajectory length and Y axis as solving time.**
>
> See the global response and attached pdf for additional results to more comprehensively evaluate the scalability w.r.t. trajectory length and problem size.
>
> We hope you will consider raising your score given we will address your editorial comments and have provided additional scalability results.

---

### Official Review · Reviewer_kJe4 · 2023-07-18

**Soundness:** 3 good
**Presentation:** 3 good
**Contribution:** 2 fair
**Rating:** 7
**Confidence:** 4

**Summary:**

Differentiating through optimal control problems to learn various components, such as the dynamics model or cost function, is a promising method for inverse reinforcement learning (IRL) or incorporating more structure in learned control policies. The central component of these approaches is to differentiate through optimality conditions, such as the KKT conditions or Pontryagin's maximum principle (PMP). Methods which differentiate through the PMP scale linearly with the planning horizon by constructing an auxiliary control system solved in the backward pass. Prior work has argued that this is not the case for methods which differentiate through the KKT conditions due to a large matrix inversion. Instead, this paper shows that this is not true if one properly accounts for the block structure and sparsity patterns of the matrices. The authors show that their approach also scales linearly with horizon and has more opportunity for parallelism to enable faster gradient computation. They compare to Pontryagin Differentiable Programming (PDP), a method derived from PMP, on a number of standard benchmark problems in the case of inverse reinforcement learning. Both methods perform similarly most of the time, with PDP sometimes failing, potentially due to issues with numerical stability. They also show this gap widens when inequality constraints are introduced, with their method yielding significantly better gradients and imitation loss.

**Strengths:**

- Improving the scalability of implicit differentiation for learning parameters of optimal control problems is an important problem with applications in IRL, system identification, and structured feedback policy classes for reinforcement and imitation learning.
- The paper is well organized and clearly written. It does a good job explaining the novelty and results and provides enough information to support its claims.
- This paper shows that computing gradients through the KKT conditions for general optimal control problems with inequality and equality constraints can also scale linearly with horizon when we properly account for the matrix structure. Additionally, they show that methods which differentiate through the PMP conditions are equivalent to their approach, only differing in the use of a recursive rule for gradient computation.
- Parts of the gradient computation in their method is parallel across time steps, unlike prior work which is entirely sequential. This allows them to compute gradients and vector-Jacobian products much more quickly.
- Gradients appear to be more stable, especially in the case of inequality constraints, compared to PDP and its extensions. The computation time for the backwards pass is also significantly faster for the proposed method over PDP. This may enable scalability to longer horizon problems.
- Unlike prior methods which efficiently differentiate through the KKT conditions, such as [2], they do not rely on differentiating through an approximation of the non-convex problem. Instead, they compute gradients through the original problem, which may have benefits in terms of gradient quality.

**Weaknesses:**

- The central goal of the paper is to improve the scalability and numerical stability of gradient computation for long horizons. However, there is no evaluation of runtime or gradient stability across different horizon lengths. Instead, the benchmarks use a fixed horizon, which is of moderate length. It would strengthen the paper to see a breakdown of how the improvements over PDP scale with the horizon length of the problem, and if these trends carry over to even longer horizons than currently considered in the paper. If PDP truly scales worse due to numerical stability issues, then its performance should get worse with longer horizons while the proposed method does not.
- There is no discussion of what solver is used in the forward pass for the experiments and how the Lagrange multipliers are found for gradient computation. Even if the paper reuses the methods from the PDP paper, the paper should still be stand-alone in that it contains these details in the appendix.
- The paper says that differentiable MPC [2] is limited to affine-quadratic systems, which is not true. By using iLQR, it is able to handle nonlinear dynamics and non-convex cost functions. The paper also shows how to incorporate box constraints on controls. However, the proposed approach is more general in that it can handle arbitrary constraints. This should be fixed in the final paper.
- This method is not the first to exploit structure in the KKT conditions to scale linearly with horizon. Despite the arguments in [18, 20], differentiable MPC [2] does not involve a large matrix inversion. Instead, it scales linearly with horizon by solving an auxiliary LQR in the backwards pass, similar in spirit to PDP in [18, 20]. While the proposed method is more general, there should be some discussion of this relationship. It would also strengthen the paper to include it as a benchmark given that it is also derived from the KKT conditions. It would especially be interesting to see how the quality of gradients and runtime of the backwards passes compare.

**Questions:**

- How does the gradient stability of the proposed method compare to PDP and its related methods as horizon increases?
- How does the proposed method fair with much longer horizons than those considered in the experiments?
- What is the solver used in the forward pass and how are the Lagrange multipliers found?

**Limitations:**

There is some discussion of how the bottleneck on speed is now in the forward pass, given that the backward pass has been significantly sped up. The speed of the forward pass will also depend on the choice of solver for the constrained optimization problem. And there is some discussion of how the opportunities for parallelism in the backward pass are limited due to the availability of solvers for general block tridiagonal linear systems.

---

> ### Author Rebuttal · Authors · 2023-08-09
>
> Thank you for your appreciation of the contributions proposed in our work. We would like to address your concerns.
>
> > **It would strengthen the paper to see a breakdown of how the improvements over PDP scale with the horizon length of the problem, and if these trends carry over to even longer horizons than currently considered in the paper.**
>
> We have provided additional insight and analysis around runtime scalability in the global response. In short, we believe we have firmly demonstrated this by evaluating both computation time and numerical stability on a synthetic example with varying $T$ up to $T=1000$.
>
> > **There is no discussion of what solver is used in the forward pass for the experiments and how the Lagrange multipliers are found for gradient computation.**
>
> We are currently using the primal dual interior point solver IPOPT which solves for optimal trajectories and provides Lagrange multipliers. We will include these details in the paper and make our paper more stand-alone. We will also mention that we can compute the Lagrange multipliers given the solution as in PDP and Safe-PDP.
>
> > **Despite the arguments in [18, 20], differentiable MPC [2] does not involve a large matrix inversion.**
>
> Thank you for mentioning this, we agree completely after revisiting the paper. We will reflect these changes in the manuscript, notably that the advantage of IDOC and Safe-PDP is around handling more general constraint functions and furthermore, that DiffMPC is very similar to PDP in that an auxiliary LQR problem is solved.
>
> Interestingly, the method proposed in DiffMPC solves the VJP directly, and as such as is a worthwhile additional comparison. However, we note that in Module 2 in the DiffMPC paper for handling non-convex problems (under Backward pass, steps 4-5), a **third-order derivative** must be computed. Specifically, this is the derivative of (w.r.t. $\theta$) of the derivative of $\ell$ w.r.t. $H^n_\theta$. This will significantly affect computation time of DiffMPC for large-scale problems.
>
> While we have not had sufficient time to perform the comparison during the rebuttal period, we will attempt to perform a comparison during the discussion period.

---

> > ### Comment · Reviewer_kJe4 · 2023-08-20
> >
> > Thank you for your response! I believe that the additional experiments and details will strengthen the paper. If the additional comparisons to DiffMPC could make the final paper, including computation time and performance, that would be great but not necessary in my opinion. I will adjust my score accordingly.

---

### Official Review · Reviewer_ZYtx · 2023-08-01

**Soundness:** 3 good
**Presentation:** 2 fair
**Contribution:** 2 fair
**Rating:** 4
**Confidence:** 3

**Summary:**

The paper introduces a method for calculating analytic trajectory gradients in constrained optimal control problems using implicit differentiation, with following contributions:
* Shows that computation of these derivatives can be linear in trajectory time-steps, utilizing the structure in the matrices in the gradient computation.
* Shows how to parallelize the gradient computation for reduced overall computation time and better numerical stability.
* Shows direct computation of vector-jacobian products for find optimal trajectories with some outer loss.

The paper utilizes results from previous works [15,18] and uses insights from the those results to build incremental contributions. Though incremental, the insights can be practically useful.

**Strengths:**

Strenghts:
The contribution of the method can be practically useful for trajectory gradient methods allowing for faster computation.
Originality - The insights and the contributions for the paper are original to best of my knowledge. Using block sparse structure of gradient computation for linear time complexity in time-steps is the main original idea.
Quality - The quality of contribution is moderate. Since the paper is mainly is analytical, built on insights based on previous results and the speed-up is only 2X.
Clarity - I believe the writing of the paper can be more clear, especially section 3 onwards. Subsection 4.2 needs more work to be easy to parse.
Significance - Significance is moderate. Since the speed is only 2X, and also in figure 3, the improvements over baselines are minor in 3 out of 4 environments. In Figure 4 as well, the final final imitation loss seems very close for different methods, though the gradient quality indeed seems better.

**Weaknesses:**

Major Weaknesses:
1. The paper lacks comparison on other experimental setting such as sysID and control from the baseline paper [18].
2. It seems the relevance for the paper is mostly in the setting where trajectory gradients wrt to model parameters or objective parameters is required such as LfD. For a holistic comparison, there should be benchmarking wrt to current important methods specialized for SysID (model learning in MBRL) or inverse RL.
Minor Weaknesses:
1. Improve the legends in figure 3, and enlarge figure 1(a). In general, all the figures needs improvement.

**Questions:**

I have a few questions related to the significance of this work. If you can help me with following questions to better place the paper in the field of optimal control and RL.
1. Is the method introduced only limited to LfD setting or are there any other setting in RL or optimal control this method can be used for?
2. Can you place this work with respect to model-based/model-free/inverse RL methods? i.e if this method or any part of this method can be used for policy learning or planning or model learning?

**Limitations:**

Limitations:
* No clear experiments about how the method scales with more challenging and complex environments such as DM control suit, etc.

---

> ### Author Rebuttal · Authors · 2023-08-09
>
> Thank you for your review. We will gladly address editorial comments around legends and writing. In addition, we will address the concerns you have raised under both **Strengths** and **Weaknesses**.
>
> > **Quality - The quality of contribution is moderate. Since the paper is mainly is analytical, built on insights based on previous results and the speed-up is only 2X.**
>
> Whilst acknowledging that our contribution is largely analytical, we note that it holds for a very general class of problems with the state-of-the-art split over two papers (PDP and Safe-PDP). Furthermore, we believe clarifying misconceptions around the implicit function theorem (see our response to reviewer tsv7) is of value to both the learning and control communities.
>
> We hope our new numerical experiments in the global response changes your opinion around the relative speed up!
>
> > **Significance - Significance is moderate. Since the speed is only 2X, and also in figure 3, the improvements over baselines are minor in 3 out of 4 environments.**
>
> Furthermore, our extra numerical results presented in the global response now clearly demonstrate a significant numerical performance improvement over the state-of-the-art on large-scale challenging problems (now an approximate 10x speedup, with further gains possible for larger problem sizes).
>
> Similarly, our extra numerical results further emphasise the significant gains in gradient quality over state-of-the-art achieved by differentiating through problems with inequality constraints without resorting to approximate log-barrier problems. These stability improvements are on top of the observed speedups.
>
> > **In Figure 4 as well, the final final imitation loss seems very close for different methods, though the gradient quality indeed seems better.**
>
> Additional analysis exploring the benefits of avoiding the use of log-barrier approximations inherent to Safe-PDP is provided in the global response. Importantly, this analysis extends beyond imitation loss, highlighting that avoiding the log-barrier allows for the correct recovery of the true constraints and system parameters, which cannot be observed from plots of imitation loss alone.
>
> > **It seems the relevance for the paper is mostly in the setting where trajectory gradients wrt to model parameters or objective parameters is required such as LfD. For a holistic comparison, there should be benchmarking wrt to current important methods specialized for SysID (model learning in MBRL) or inverse RL.**
>
> We believe the CIOC problem we evaluate on where the dynamics and the cost are jointly learned is more difficult than just the SysID and the IRL tasks proposed in the PDP paper. In fact, the gradient computations which arise from the SysID task amounts to just backpropagating through the dynamics equations. This renders differentiating through the KKT/PMP conditions completely unnecessary. This observation also holds for the control and planning tasks. However, we are happy to add additional results to the appendix around IRL (known dynamics) and SysID (known cost) for completeness.
>
> > **Can you place this work with respect to model-based/model-free/inverse RL methods? i.e if this method or any part of this method can be used for policy learning or planning or model learning?**
>
> Thank you for suggesting this. LfD is the most obvious task to apply differentiable COC, however we have added some additional commentary in the global response. We believe our method can be used anywhere a trajectory derivative is required. For example, we provide a reference to an RL work which learns an implicit policy, defined as the solution to an optimal control problem. We hope our method will further encourage and enable the learning of these so-called implicit policies.

---

> > ### Comment · Reviewer_ZYtx · 2023-08-11
> >
> > I have read the rebuttal. Thanks for the comments and added experiments. I believe that added experiments adds extra value to the paper and strengthens it. Though I believe overall - the presentation, writing, method writing, experiments - need extra work. I also think the contributions of method are more of computational insights i.e. one of the main result/contribution of the paper - real fast computation of derivatives in IDOC VJP - is built from the insight about block structure in results derived in previous work and then doing multiplication from left-to-right. Having said that, I believe that these results can be helpful for the community and should be disseminated. But i will encourage the authors to revise the paper with all the mentioned feedback from reviewers to have more appealing article and submit again.

---

> > > ### Author Response · Authors · 2023-08-16
> > > **Thank you**
> > >
> > > Thank you for your response. We appreciate you supporting our work and for agreeing that our results can be helpful for the community and should be disseminated. We’re hoping to do this sooner rather than later and will certainly incorporate all reviewer feedback in subsequent versions of our paper.

---

### Author Rebuttal · Authors · 2023-08-09

# Global Response Overview

We thank the reviewers for providing detailed comments and feedback on our work. We are pleased the reviewers appreciated the computational and numerical performance improvements compared to existing state-of-the-art offered by our use of (structured) implicit differentiation and vector-Jacobian products for optimal control (IDOC and IDOC VJP, respectively). We address a number of common concerns raised by multiple reviewers in a series of posts. Specifically, we address

* Additional results around speed up and numerical stability
* Emphasis around handling problems with inequality constraints
* Making the paper more stand-alone

and leave responses to specific comments from individual reviewers as a reply to their respective posts.

## Additional Numerical Results

We constructed a large, synthetic example to more comprehensively test the computation time saving and numerical stability of IDOC. We sample the blocks, comprising terms in Eq. 4 in the original submission, randomly such that they are 1) large, and 2) poorly conditioned to test the effect of parallelisation and numerical stability, respectively. In summary, our results indicate:

* IDOC, IDOC VJP and PDP **scales linearly** w.r.t. horizon length $T$ and is approx. 10x faster for $p=2000$.
* IDOC VJP scales an **order of magnitude** better than both IDOC full and PDP w.r.t. the number of parameters $p$.
* IDOC is **more numerically stable** than PDP and degrades more gracefully as the numerical conditioning of the blocks deteriorates.

See the 1 page response pdf for figures (Fig. 1) and more experimental details.

## Importance of Handling Inequality Constrained Problems

We would like to emphasize the significance of the improvement IDOC affords over Safe-PDP for differentiating through optimal control problems with inequality constraints. We acknowledge that the log-barrier approach proposed in Safe-PDP yields similar imitation loss IDOC. However, there are additional benefits to avoiding a log-barrier approach and differentiating through the constrained problem.

* Recovery of the correct constraint function parameters when learning from demonstration data (see Fig. 2 in the response pdf).
* Avoids necessitating initializing the forward pass with a feasible trajectory (otherwise the log-barrier function is undefined).

## Difficulty of Experiments

Some reviewers questioned the difficulty of the experiments. While we agree that scaling up these control problems to more difficult settings (e.g., the DM control suite) is of interest to the community generally, our particular contribution is largely analytical and relates only to the gradient computation. We believe the performance gain of IDOC over PDP/Safe-PDP are already significant, especially when considering the simple nature of the problems we evaluate on. We hope the additional challenging synthetic experiments sufficiently substantiate our claims around numerical stability and parallelization.

## Improving the Stand-Alone Quality of the Paper

We will make the following changes to our paper to make it more stand-alone.

* Describe for each experiment the dynamics equations and cost function parameterization in the appendix.
* Describe the forward solver used (IPOPT) and associated hyperparameters during learning.
* Add more visualizations to the appendix around the experiments.
* Better place the work in the broader RL literature, rather than simply comparing against PDP/Safe-PDP. We believe IDOC is well suited to model-based RL approaches such as the ones proposed in [1], which solve an OC problem (described in Equation 2 in our submission) to produce actions (implicit policy). We hope our work will encourage future work around this direction.

### References

[1] Romero, A., Song, Y. and Scaramuzza, D. (2023). *Actor-Critic Model Predictive Control.* arXiv preprint arXiv:2306.09852.

---

### Decision · Program_Chairs · 2023-09-21

**Decision:**

Accept (poster)

**Comment:**

Thank you for your submission and active engagement throughout the review period.
After the discussion period, the reviewers and I are in agreement that this is
a reasonable advancement to implicit differentiation for optimal control and
the methodology and experimental results to Pontryagin differentiable programming
clearly demonstrate the advantages.